# Prevalence of mortality caused by injuries at Livingstone University Hospital, Zambia. A retrospective cross-sectional study

**Lukundo Siame**[1]*, **Malan Malumani**[1,2], **Chiyeñu O. R. Kaseya**[2], **Sergiy Ivashchenko**[1,2], **Leah Nombwende**[1], **Sepiso K. Masenga**[1], **Benson M. Hamooya**[1], **Michelo Haluuma Miyoba**[1,2]

1 School of Medicine and Health Sciences, Mulungushi University, Livingstone, Zambia, 2 Livingstone University Teaching Hospital, Livingstone, Zambia

* lukundosiame23@gmail.com

**Data Availability Statement:** All relevant data are within the manuscript and its Supporting information files.

## Abstract

### Background

Trauma is a major global public health issue, with an annual death toll of approximately 5 million, disproportionately affecting low- and middle-income countries. Zambia bears a significant burden of trauma-related mortalities, contributing to 7% of all annual deaths and 1 in 5 premature deaths in the country. Despite the significant burden of trauma in our country, few studies have been conducted, with most focusing on high-population centers, and there is a lack of epidemiological data on trauma-related deaths in our region. Therefore, our aim was to estimate the proportion of deaths caused by injuries at Livingstone University Teaching Hospital, a tertiary hospital located in Zambia's southern province.

### Methods

We conducted a retrospective cross-sectional study from June 22, 2020, to June 22, 2021, among 956 individuals from 1 month old (29 days of age) to 100 years. Demographic and clinical data were collected from patient's records from Accident and Emergency department. Data analysis included descriptive statistics, chi square, mann-whitney test and multivariable logistic using forward stepwise generalized linear model equations (GLM) to identified factors associated with mortality, with a significance level set at $p < 0.05$. Data were analyzed using STATA version 15.

### Results

Among the study participants, the median age was 26 years (interquartile range (IQR) 15, 37) and the majority were males (74.2%, n = 709). Prevalence of mortality was 1.0% (n = 10). The deaths were caused by burns (60%, n = 6), violence (30%, n = 3), and traffic accidents (10%, n = 1). Among those who died, the majority of the trauma occurred at home (90%, n = 9), followed by road (10%, n = 1) and were as a result of burns (60%, n = 6) and community violence (30%, n = 3). Survivors had significantly higher treatment costs (ZMK 9,837 vs. ZMK 6,037, p<0.005). Having burns (AOR: 1.06, 95% CI: 1.05, 1.09, p< 0.001)

**Funding:** The author(s) received no specific funding for this work.

**Competing interests:** The authors have declared that no competing interests exist.

and hospital stay of one day (AOR: 1.04, 95% CI: 1.02, 1.05, p< 0.001) was positively associated with mortality, while hospital stay of more than five days (AOR: 0.98, 95% CI: 0.96, 0.99, p = 0.002) was negatively associated with mortality.

## Conclusion

The prevalence of death due to trauma was relatively low, with the majority experiencing multiple traumas. Burns were the most common cause and were associated with death, occurring within a day of hospitalization. The findings underscore the need for targeted preventive measures, improved access to quality emergency trauma care, and rehabilitation services, especially among patients with burns.

## Introduction

Trauma is a global public health problem affecting every country, particularly low and middle-income countries [1]. Despite it being a huge burden on society, it is often ignored in modern society [2]. According to the World Health Organization (WHO), approximately 5 million people die yearly or about 16,000 people die daily, from trauma globally [3]. This is nearly 1.7 times the number of deaths because of human immunodeficiency virus (HIV) acquired immunodeficiency syndrome (AIDS), tuberculosis (TB) and malaria combined [1]. Over a quarter of these fatalities are due to road traffic accidents (RTAs), primarily in Africa and other low-income countries [2], and disproportionately impact economically active young people [4].

Zambia, a sub-Saharan African country, bears a significant burden of trauma-related deaths, accounting for approximately 20% of premature deaths and 7% of all annual deaths [5]. The financial burden of injury costs falls heavily on individuals, with out-of-pocket payments accounting for a significant proportion [6]. As Zambia is undergoing rapid economic growth and urbanization, with 50% of the population projected to live in urban areas by 2035, the burden of injuries is expected to rise, mainly driven by increased motorization and poor urban development, which include poor road infrastructure and a lack of planned space for cyclists and pedestrians [5, 7]. In addition there is lack of public health awareness programmes such as mandatory use of helmets, seatbelts and child restraints, vehicle inspection services, and lack of research focused on injury are contributing factors [8, 9]. Furthermore, there is limited pre-hospital and in-hospital service for injured patients in the already overburdened healthcare system [1, 7].

Despite the significant burden of traumatic injuries in Zambia, there is limited information on mortalities associated with traumatic injuries. Therefore, this study aimed to estimate the magnitude of injury associated mortalities at Livingstone University Hospital, Southern Province, Zambia.

## Methods

### Study setting

This study conducted was conducted at the Accident & Emergency (A&E) Department at the Livingstone University Teaching Hospital (LUTH), the third-level hospital and the largest government hospital in Southern Province and serves as a referral center for patients from neighboring western province, Zambia. LUTH has about 325 bed spaces, and the Accident and Emergency has 16 beds dedicated to emergencies and 6 intensive care unit beds located in the

Surgical Ward of the hospital. The department receives about 30 to 40 trauma patients monthly. Four surgeons, including two general surgeons, one neurosurgeon, one orthopedic surgeon, one anesthesiologist, and an intensive specialist, lead A & E department, with support from senior resident medical officers, junior resident medical officers, emergency nurses and sonographers.

## Study design and data

This study was a retrospective cross section study that abstracted data from the A & E of all Medical records of patients who sustained any type of physical injury, as determined by a treating doctor from 1$^{St}$ January 2018 to 31$^{st}$ December 2019. The medical records record socio-demographic details (age, gender, place of residence, and employment status), injury-related information (date and time of injury, cause of injury, injury site, and anatomical location of injury), and treatment-related data (investigations ordered, such as CT scans, X-rays, and laboratory tests; interventions performed, including blood transfusions and surgical procedures) and outcome of the patient which are categorized as discharge from A&E, hospital admission, or death.

## Eligibility and sampling method

The study enrolled all patients aged 1 month (29 days of age and 100 years of age presenting with any traumatic injury at the A & E Department. We abstracted all the files between the study period (n = 4067) and exclude all participants without evidence of traumatic injury (n = 3111), see Fig 1. All records were entered into research electronic data capture (Kobotool box application), a data management tool.

## Study variables

In this study, death was the dependent variable, while age, sex, address, injury setting, cause of injury, type of injury, intervention received, length of stay at the hospital, and estimated cost were independent Variables. In this study, multiple area was defined as the presence of injury to more than one body area or system e.g. upper limbs and lower limbs of Head and abdomen. Community violence, referred to deliberate acts of interpersonal violence in public spaces by individuals not intimately related to the victim. The study also classified injury setting, home meant any injury (falls, burns, or interpersonal violence) which occurred within a residential environment like a house or apartment; Road setting meant any injuries resulting from vehicle accident involving drivers, passengers, cyclists, or pedestrians; Work setting meant any injuries which happens while performing job related tasks; School meant any injuries which happens. Non- surgery are patients who received conservative treatment. The cost of patient care was assessed based on direct costs in Zambian Kwacha (ZMK), which included dressings, Foley catheters, nasogastric tubes, meals, bed space, nursing care, drugs, investigations, and surgery. Indirect costs related to management were not included in the analysis.

## Data collection and study procedures

Demographic and clinical data were abstracted from patient files. The data were collected using the Kobo Toolbox application by two (2) trained research assistants (one nurse and one medical doctor). Subsequently, the data was checked for accuracy and duplicate entries, with any duplicates being removed by the senior abstractor (surgeon). This data was collected between October 2, 2020, and February 21, 2021.

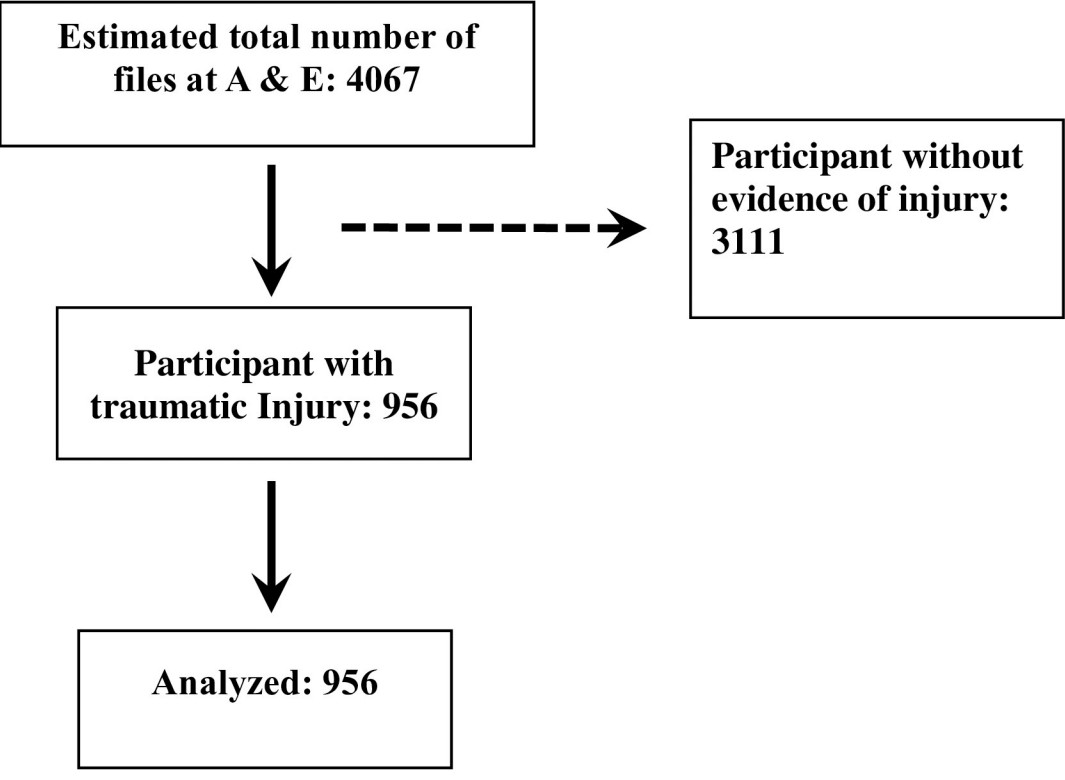

**Fig 1. Flowchart of screened and eligible files.**

## Statistical analysis

Data were first abstracted from paper-based medical records, then into an online tool (a Kobo-tool application). Data cleaning was done in Excel, coded, and then exported to Stata version 15 for analysis. We tested the normality of continuous variables using Shapiro-Wilk's test. We then used descriptive statistics, frequencies and percentages to summarize categorical variables, and the median (interquartile range) for continuous variables. The chi-square test was used to establish a statistical difference between the outcome and other categorical variables, while Mann-Whitney test was used for continuous variables. Multivariable logistic regression with forward stepwise generalized linear model equations (GLM) was used to identify the factors associated with mortality. The statistical significance level was set at $p < 0.05$.

## Ethics

Ethical clearance was obtained from Mulungushi University School of Medicine and Health Sciences Research Ethics Committee (reference number SMHS/MU2/2020-28) on 22$^{nd}$ June, 2020. All data that underwent analysis were de-identified to protect confidentiality. No patient-specific details that can be used to identify the participant was abstracted or recorded in the data collection process, thus preventing the identification of any participant during or after data collection. Written consent or assent for individual participants was not required and was therefore waived, as the data collected were secondary in nature.

In reporting this observational study, we followed the guidelines outlined by the Strengthening the Reporting of Observational Studies in Epidemiology (STROBE), as outlined in S1 Checklist.

## Results

### General characteristics

The study comprised 956 trauma injury patients, of whom 709 (74.2%) were male and the median age was 26 years old (interquartile range (IQR) 15, 37). Most of the trauma occurred among urban residents (69.1%, n = 637). The median (IQR) cost for trauma was ZMK9, 837 (IQR: 3437, 14637). A higher proportion of injuries occurred at home (59.5%, n = 566), followed by on the road (32.3%, n = 306). The most common cause of trauma was road traffic accidents (32.1%, n = 306) while community violence (24.9%, n = 237) followed. The highest number of patients had multiple trauma (30.3%, n = 290), followed by head injury (28.2%, n = 270), then lower limb trauma (22.5%, n = 215). The most frequent intervention was minor surgery (58.5%, n = 559). Three hundred and thirty-six (35.2%) participants were admitted to the hospital for more than five days, followed 153 for one day (16.0%) and 191 by less than 24 hours (20.0%), Table 1.

### Prevalence of death and its relationship with other study variables

The prevalence of mortality was 1.0% (n = 10; 95% confidence interval 0.5, 1.9). Participants who died were younger than those who survived, 7 years vs. 26 years; p = 0.0991. The participants who died incurred lower costs in comparison to those who survived (ZMK 6037 vs. ZMK9837, p = 0.049). The majority of individuals who died were from urban areas (66.3%, n = 6), and injuries occurred at home (90.0%, n = 9) and were as a result of burns (60.0%, n = 6), and most of them had multiple (70%, n = 7) and head (30%, n = 3) as the site of injuries. A higher proportion of participants who died were hospitalized for a day (70.0%, n = 7). Additionally, most of the participants who died had undergone minor surgery, (60.0%, n = 6), Table 2.

### Regression analysis of factors associated with mortality

Table 3 shows a multivariable analysis of the factors associated with mortality. In the multivariable analysis, participants with burn injuries had significantly higher odds of dying 1.06 times compared to those without. Hospital admissions lasting one day increased the odds of death by 1.06 times, while stays lasting more than 5 days reduced the odds of death by 0.98 times.

## Discussion

Our study aimed to determine the prevalence of injury associated mortalities at Livingstone University Hospital, Southern Province, Zambia. We found that most of the deaths occurred predominantly in young males, aligning with studies conducted in Malawi and Nigeria, which also showed a predominance of young males [10, 11]. The possible explanation for this trend in our setting could be that most of the young men are more likely to be involved in violence due to the high consumption of alcohol and most of the men are over-represented in hazardous occupations such as transportation, security guard, construction, and mining [12–15]. These factors are similar to other studies that have shown that this demographic is more prone to involvement in violence, conflict, and dangerous occupations [16, 17]. Additionally, societal expectations shaped by traditional masculinity norms potentially exacerbating the issue [11, 18, 19].

We found that patients who died incurred lower management costs compared to those who survived, [20]. possibly due to shorter hospital stays. Additionally, we discovered that there were few studies on the direct cost of trauma management, and those available utilized different models for these calculations. However, according to Wesson et al. (2014), the average cost of managing trauma patients in low and middle-income countries is estimated to be between

**Table 1. Descriptive statistics: Demographic and clinical characteristics of trauma patients.**

| Variable | Median (IQR) | Frequency | Percentage |
|---|---|---|---|
| Age in years, | 26 (15, 37) | | |
| Estimated cost (ZMK), | 9,837 (3437,14637) | | |
| **Sex** | | | |
| Female | | 247 | 25.8 |
| Male | | 709 | 74.2 |
| **Address** | | | |
| Urban | | 637 | 69.1 |
| Rural | | 275 | 29.9 |
| Non-Zambian | | 9 | 1 |
| **Injury Setting** | | | |
| Home | | 566 | 59.2 |
| Road | | 310 | 32.4 |
| Work | | 56 | 5.9 |
| School | | 24 | 2.5 |
| **Causes of injury** | | | |
| Community Violence | | 237 | 24.9 |
| Road traffic | | 306 | 32.1 |
| Burns | | 108 | 11.3 |
| Falls | | 137 | 14.4 |
| Animal attack | | 104 | 10.9 |
| Others | | 61 | 6.4 |
| **Location of injury** | | | |
| Head trauma | | 270 | 28.2 |
| Abdominal trauma | | 12 | 1.3 |
| Spinal trauma | | 18 | 1.9 |
| Pelvic and perineum trauma | | 18 | 1.9 |
| Chest trauma | | 42 | 4.4 |
| Lower limb trauma | | 215 | 22.5 |
| Upper limb trauma | | 91 | 9.5 |
| Multiple areas | | 290 | 30.3 |
| **Intervention** | | | |
| Major-Surgery | | 266 | 27.8 |
| Minor-Surgery | | 559 | 58.5 |
| Non surgery | | 131 | 13.7 |
| **Duration of stay at the Hospital** | | | |
| <24 hours | | 191 | 20 |
| 1 day | | 153 | 16 |
| 2 day | | 124 | 13 |
| 3 day | | 93 | 9.7 |
| 4 day | | 58 | 6.1 |
| >5 day | | 336 | 35.2 |

**Abbreviation**: IQR (interquartile range); ZMK Zambian kwacha

US$14 and US$17,400 (currently equivalent to 378 and 469, 800 Zambian kwacha) [20]. The cost of managing patients who died fell within this range. This high cost suggests a substantial economic burden of injury, emphasizing the need to invest in injury prevention, more especially in resource-limited settings.

**Table 2. Univariate analysis with factors associated with death.**

| Characteristics | Alive, n = 946 (99.0%) | Dead, n = 10 (1.0%) | P value |
|---|---|---|---|
| **Age in years*** | 26 (15, 37) | 7 (5, 28) | 0.0991 |
| **Estimated cost in Zambian Kwacha*** | 9837 (3437, 14637) | 6037 (1837, 9837) | **0.049** |
| **Sex**** | | | 0.762 |
| Male | 702 (99%) | 7 (0.99%) | |
| Female | 244 (98.8%) | 3 (1.21%) | |
| **Address**** | | | 0.951 |
| Urban | 630 (69.1%) | 7 (70.0%) | |
| Rural | 272 (29.8%) | 3 (30.0%) | |
| Non-Zambian | 9 (0.9%) | 0 (0%) | |
| **Injury Setting**** | | | 0.376 |
| Home | 557 (58.9%) | 9 (90.0%) | |
| Road | 309 (32.6) | 1 (10.0%) | |
| Work | 56 (5.9%) | 0 (0%) | |
| School | 24 (2.5%) | 0 (0%) | |
| **Causes of injury**** | | | **0.002** |
| Community violence | 234(24.8%) | 3 (30.0%) | |
| Road traffic | 305 (32.3%) | 1(10.0%) | |
| Burns | 102(10.8%) | 6(60.0%) | |
| Falls | 137 (14.5%) | 0(0%) | |
| Animal attack | 104 (11.0%) | 0(0%) | |
| Others | 61 (6.5%) | 0 (0%) | |
| **Location of injury**** | | | 0.286 |
| Head | 267 (28.2%) | 3(30.0%) | |
| Abdominal | 12(1.3%) | 0(0%) | |
| Spinal trauma | 18 (1.9) | 0(0) | |
| Pelvic and perineum trauma | 18(1.9%) | 0 (0%) | |
| Chest trauma | 42 (4.2%) | 0(0%) | |
| Lower limb trauma | 215 (22.7%) | 0 (0%) | |
| Upper limb trauma | 91 (9.6%) | 0 (0%) | |
| Multiple areas | 283(29.9%) | 7 (70.0%) | |
| **Intervention**** | | | 0.181 |
| Major-Surgery | 265 (28.0%) | 1 (10.0%) | |
| Minor-Surgery | 553 (58.5%) | 6 (60.0%) | |
| Non-surgery | 128 (13.5%) | 3 (30.0%) | |
| **Length of stay at the hospital**** | | | **< 0.001** |
| <1 day | 191 (20.2%) | 0 (0%) | |
| 1 day | 146 (15.5%) | 7 (70.0%) | |
| 2 day | 122(12.9%) | 2 (20.0%) | |
| 3 day | 92(9.7%) | 1 (10.0%) | |
| 4 day | 58(6.1%) | 0(0.0%) | |
| > 5 day | 336 (35.6%) | 0(0%) | |

Abbreviation:

* data presented as median (interquartile range),

** data presented as frequency (%),; ZMK: Zambian kwacha, others include injuries like work accident, electrical electrocution, pricks from sticks

**Table 3. Multivariable logistic regression of factors associated with mortality.**

| Variable | AOR (95% Cl) | P-value |
|---|---|---|
| **Cause of Injury** | | |
| Community violence | Ref | |
| Road traffic | - | - |
| Falls | - | - |
| Animal attack | - | - |
| Others | - | - |
| Burns | 1.06 (1.05, 1.09) | < **0.001** |
| **Length of stay at the hospital** | | |
| <1 day | Ref | |
| 1 day | 1.04 (1.02, 1.05) | < **0.001** |
| 2 day | - | - |
| 3 day | - | - |
| 4 day | - | - |
| > 5 day | 0.98 (0.96, 0.99) | **0.002** |
| **Intervention** | | |
| Major-Surgery | Ref | |
| Non-surgery | - | - |
| Minor-Surgery | 0.99 (0.98,1.00) | 0.091 |

**Abbreviation**: AOR: adjusted odds ratio, 95% Cl: 95% confidence interval, others include injuries like work accident, electrical electrocution, pricks from sticks

This study revealed that the majority of individuals who died were from urban areas. This finding is similar to a study conducted across multiple sub-Saharan African (SSA) regions, which showed that most injuries occurred in urban areas, particularly around the home setting rather than on roads [21–23]. However, this is in direct contradiction with other studies, such as those carried out by Diamond et al. (2018) and Siedenburg et al. (2014), which indicated that the road setting is the most common setting for injuries in SSA [7, 24]. One possible explanation could be attributed to the fact that Livingstone is not a mega-city, unlike Lusaka, for example. This is supported by a study carried out by Cabrera-Arnau et al. (2020), which showed that places with smaller population sizes tend to have fewer road accidents [25].

Burns injuries accounted for the majority of recorded deaths, aligning with studies conducted in Nigeria, which identified burns as the most common cause of mortalities [11]. Possible factors for this mortality rate among patients with burns could be as a result of the hospital not having a dedicated unit for burns with specialized equipment and specialized personnel. Other factors include delayed referral to the hospital, most burns are more than 20% and most burns are infected and mixed and deep burns. Forbinake et al (2020) in Cameroon and Mulatu et al (2022) in Ethiopia, found that the factors contributing to these fatalities are most likely due to the severity of the burn injury, age, type of burn, and insufficient early resuscitative measures in many low-income countries like Zambia [26, 27].

In this study, the majority of deaths occurred within the first day of hospitalization, while patients who survived beyond five days generally had better outcomes, These finding are consistent with studies by Valdez et al. (2016) and Bardes et al. (2018) [28, 29]. There are two primary causes for why this the case, mostly patients who pass away prematurely often have non-survivable injuries that would ultimately lead to their early demise, irrespective of the resources they receive [28]. Another group of early-death patients may have survivable injuries, but they

do not receive the immediate and exceptional medical care necessary to save their lives as a results of absences of cutting edge technology equipment and human resources which is a challenge in our setting [28].

The overall mortality rate of trauma in Africa varies. A study in rural Kenya and Lusaka, Zambia, reported rates of 3.5% and 3%, respectively [7, 30], while studies conducted at tertiary hospitals in Malawi [10] and South Africa [31] found rates of 4.2%. Conversely, our study observed a lower mortality rate of 1%. The low rate of death in our hospital can be attributed to good trauma management despite not having specialized equipment and personnel. However, it has also been shown that there is underreporting of many trauma-related deaths as many fatalities are taken directly to the hospital mortuary, thus not documented in the hospital registry [32].

The studies had some limitations, one of the most significant was the inability to capture all the incident data on burns, because deaths that occur outside the hospital are not recorded as part of the hospital statistics from within Livingstone district. Despite the limitation, this study was able to quantify and describe the characteristics of trauma-related deaths in a resource-limited setting.

## Conclusion

Our study revealed a low prevalence of trauma-related deaths, with burns as the leading cause, overwhelmingly occurring within the home environment. The high cost of managing these patients exposes a critical need for a proactive approach. We must prioritize preventive measures, enhance access to high-quality emergency trauma care, and bolster rehabilitation services, particularly for burn victims.

## Supporting information

**S1 Checklist. STROBE statement—Checklist of items that should be included in reports of** *cross-sectional studies.*
(DOCX)

**S1 Dataset.**
(XLSX)

## Acknowledgments

We extend our sincere gratitude to the Hand group at Mulungushi University and the Senior Medical Superintendent's office at Livingstone University Teaching Hospital for their instrumental support and unwavering enthusiasm throughout our research journey.

## Author Contributions

**Conceptualization:** Lukundo Siame, Benson M. Hamooya, Michelo Haluuma Miyoba.

**Data curation:** Lukundo Siame, Benson M. Hamooya.

**Formal analysis:** Lukundo Siame, Benson M. Hamooya, Michelo Haluuma Miyoba.

**Methodology:** Benson M. Hamooya, Michelo Haluuma Miyoba.

**Resources:** Lukundo Siame.

**Supervision:** Benson M. Hamooya, Michelo Haluuma Miyoba.

**Visualization:** Malan Malumani, Chiyeñu O. R. Kaseya, Sergiy Ivashchenko, Sepiso K. Masenga, Benson M. Hamooya.

**Writing – original draft:** Lukundo Siame, Benson M. Hamooya, Michelo Haluuma Miyoba.

**Writing – review & editing:** Malan Malumani, Chiyeñu O. R. Kaseya, Sergiy Ivashchenko, Leah Nombwende, Sepiso K. Masenga, Benson M. Hamooya, Michelo Haluuma Miyoba.

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
