## [Decision Letter · Decision Letter 0]

10 Sep 2024

PONE-D-24-11129Mortality among Trauma Patients at Livingstone University Teaching Hospital: A Retrospective Cross-Sectional StudyPLOS ONE

Dear Dr. Siame,

Thank you for submitting your manuscript to PLOS ONE. After careful consideration, we feel that it has merit but does not fully meet PLOS ONE’s publication criteria as it currently stands. Therefore, we invite you to submit a revised version of the manuscript that addresses the points raised during the review process.

We look forward to receiving your revised manuscript.

Kind regards,

Rayan Jafnan Alharbi, Ph.D

Academic Editor

PLOS ONE

Journal Requirements:

Additional Editor Comments:

Thank you for submitting your work to PLOS ONE journal. I invite you to revise the manuscript, taking into consideration the reviewers' and editor's comments as described below.

Introduction:

Please provide an overview of the existing literature on the burden of traumatic injuries in Zambia, highlighting key findings from previous studies. Emphasize how the current research adds new insights or addresses gaps in the literature to differentiate it from earlier work.

Method:

Data from Accident & Emergency Department File Records:

Please provide more detailed information about the data extracted from the Accident & Emergency Department records. Specifically, how many records were available initially, and how was the sample size narrowed down to the final 956 patients? Clarify the inclusion/exclusion criteria and sampling methods used. Also, describe the types of variables typically recorded in these files—are they part of a trauma registry? Additionally, explain who was involved in the data collection process (e.g., trained medical personnel, research assistants).

Livingstone University Teaching Hospital Information:

Include more detailed information about the Livingstone University Teaching Hospital. Specifically, mention the hospital's total bed capacity, the number of beds in the Emergency Department (ED), and an estimate of the total number of trauma patients seen by the hospital, either monthly or annually.

Combining Sections:

Please remove the subtitle "Operational Definition" and merge its contents into a single paragraph under the "Data Collection and Study Procedures" section for a more streamlined presentation.

Extending Statistical Analysis:

Expand the "Statistical Analysis" section by providing more details about the statistical tests, methods, and software used. Describe the analysis process more comprehensively, including how different variables were analyzed and what criteria were used to assess statistical significance.

Results:

Please extend the second part of the results section, particularly focusing on Table 2, by providing more detailed analysis and interpretation of the data presented. This should include insights into key variables, statistical significance, and any notable trends observed in the study.

Reviewers' comments:

Reviewer's Responses to Questions

**Comments to the Author**

1. Is the manuscript technically sound, and do the data support the conclusions?

Reviewer #1: Yes

Reviewer #2: Yes

2. Has the statistical analysis been performed appropriately and rigorously? 

Reviewer #1: Yes

Reviewer #2: Yes

3. Have the authors made all data underlying the findings in their manuscript fully available?

Reviewer #1: Yes

Reviewer #2: No

4. Is the manuscript presented in an intelligible fashion and written in standard English?

Reviewer #1: Yes

Reviewer #2: No

5. Review Comments to the Author

Reviewer #1: The authors present a manuscript describing an estimate of the proportion of deaths among trauma patients at a hospital in Zambia. Overall, the manuscript will make a contribution to the field. A few suggestions to improve clarity:

Methods:

page 3: Operational Definition: more detail is needed describing injury type examined and injury settings examined. For example, for the term "various body parts", specifics are needed on what you examined and injury course i.e.home, etc.

Discussion

Page 9: The sentence "The possible explanation for this trend in our setting ..." Do you have any evidence to support this point?

Reviewer #2: It is suggested to modify the title of the article as follows.

Prevalence of mortality caused by injuries at Livingstone University Hospital, Zambia.A Retrospective Cross-Sectional Study

Table 1 and 2 are very large, it is better to follow the format of the journal.

In Table 2, the currency is written, what does the word "m" mean, if this word is necessary, it must be defined.

Reference 2 has no year. Please correct it as fallows.

Paul B, Sinha D, Misra R, Basu M, Ray S. Physical injury: Is it inevitable or preventable? an experience from a Tertiary Care Hospital of Kolkata, West Bengal. Medical Journal of Dr. DY Patil University. 2017 Nov 1;10(6):568-72.

Reference 6 is very old, please update it.

The article needs grammar editing.

6. PLOS authors have the option to publish the peer review history of their article (what does this mean?). If published, this will include your full peer review and any attached files.

Reviewer #1: No

Reviewer #2: No

---

## [Author Response · Author response to Decision Letter 0]

18 Oct 2024

13th October, 2024

To the reviewers and Editor,

 Ref: RESPONSES TO REVIEWER’S COMMENTS

We would like to thank the reviewers for taking the time to make suggestions that have improved our manuscript. We have now made revisions to the minor comments in the manuscript and incorporated all suggestions. We now hope the current manuscript is acceptable for publication. Below are the point-by-point responses to all comments and suggestions

Editor’s comments 

Introduction:

Please provide an overview of the existing literature on the burden of traumatic injuries in Zambia, highlighting key findings from previous studies. Emphasize how the current research adds new insights or addresses gaps in the literature to differentiate it from earlier work.

Response: thank you. We have now highlighted the burden of traumatic injuries in Zambia in the introduction which are in line 50 to 54. 

Method:

Data from Accident & Emergency Department File Records:

Please provide more detailed information about the data extracted from the Accident & Emergency Department records. Specifically, how many records were available initially, and how was the sample size narrowed down to the final 956 patients? Clarify the inclusion/exclusion criteria and sampling methods used. Also, describe the types of variables typically recorded in these files—are they part of a trauma registry? Additionally, explain who was involved in the data collection process (e.g., trained medical personnel, research assistants).

Response: thank you very much. We have now highlighted how data was abstracted, sampling method and what variables are recorded in the method section.

Livingstone University Teaching Hospital Information:

Include more detailed information about the Livingstone University Teaching Hospital. Specifically, mention the hospital's total bed capacity, the number of beds in the Emergency Department (ED), and an estimate of the total number of trauma patients seen by the hospital, either monthly or annually.

Response: thank you, we have now the information about Livingstone University Teaching Hospital.in the study setting section.

Combining Sections:

Please remove the subtitle "Operational Definition" and merge its contents into a single paragraph under the "Data Collection and Study Procedures" section for a more streamlined presentation.

Response: thank you. We have combined the operational in the variable section 

Extending Statistical Analysis:

Expand the "Statistical Analysis" section by providing more details about the statistical tests, methods, and software used. Describe the analysis process more comprehensively, including how different variables were analyzed and what criteria were used to assess statistical significance.

Response: Thank you very much. We have detailed the data analysis process for this paper in the methods section 

Results:

Please extend the second part of the results section, particularly focusing on Table 2, by providing more detailed analysis and interpretation of the data presented. This should include insights into key variables, statistical significance, and any notable trends observed in the study.

Response: Thank you very much. In order to provide more insight in table 2 we have now included logistic regression in order to understand the findings in this study. 

Review Comments

Reviewer #1: 

The authors present a manuscript describing an estimate of the proportion of deaths among trauma patients at a hospital in Zambia. Overall, the manuscript will make a contribution to the field. A few suggestions to improve clarity:

Methods:

page 3: Operational Definition: more detail is needed describing injury type examined and injury settings examined. For example, for the term "various body parts", specifics are needed on what you examined and injury course i.e. Home, etc.

Response: thank you. We have merged the operation definition under the variable section and provided more clarity for various body parts by naming the exact body parts. Injuried multiple areas were defined as the presence of injury to more than one body area or system, e.g., upper and lower limbs of the head and abdomen. And course injury is given in the table of results Table 1 : Descriptive statistics: demographic and clinical characteristic. Patients with a hospital length of stay less than 24 hours were treated as outpatients and discharged from the outpatient department (OPD). Those staying for one day were admitted but released the following day.

Page 9: The sentence "The possible explanation for this trend in our setting ..." Do you have any evidence to support this point?

Response: thank you for the observation. To support this statement we have provide citations now to support the statement. 

Reviewer #2:

 It is suggested to modify the title of the article as follows.

Prevalence of mortality caused by injuries at Livingstone University Hospital, Zambia. A Retrospective Cross-Sectional Study

Response: thank you for the suggestion. We have changed the title of the manuscript. 

Table 1 and 2 are very large, it is better to follow the format of the journal.

In Table 2, the currency is written, what does the word "m" mean, if this word is necessary, it must be defined.

Response: thank you very. The tables have been redrawn. And all words defined. 

We have changed the title of the manuscript

Reference 2 has no year. Please correct it as fallows.

Paul B, Sinha D, Misra R, Basu M, Ray S. Physical injury: Is it inevitable or preventable? an experience from a Tertiary Care Hospital of Kolkata, West Bengal. Medical Journal of Dr. DY Patil University. 2017 Nov 1;10(6):568-72.

Reference 6 is very old, please update it.

Response: thank you correction. The year has been added now and reference 6 has been up dated with the latest. 

The article needs grammar editing.

Response: thank you for the observation. We have done some grammar editing 

We have revised the manuscript and addressed all concerns raised by the reviewers. We want to thank you all again for the tremendous work and time that you committed in editing our work. Our manuscript is much improved, and we are very grateful. 

Yours sincerely,

Dr. Lukundo Siame, Bsc., MBcHB. 

Junior Residence Medical officer , Livingstone University Teaching Hospital

 Msc Candidate, Mulungushi University, School of Medicine

---

## [Editor Report · Decision Letter 1]

5 Nov 2024

Prevalence of mortality caused by injuries at Livingstone University Hospital, Zambia. A Retrospective Cross-Sectional Study

PONE-D-24-11129R1

Dear Dr. Siame,

We’re pleased to inform you that your manuscript has been judged scientifically suitable for publication and will be formally accepted for publication once it meets all outstanding technical requirements.

Kind regards,

Rayan Jafnan Alharbi, Ph.D

Academic Editor

PLOS ONE
---

## [Editor Report · Acceptance letter]

12 Nov 2024

PONE-D-24-11129R1 

PLOS ONE

Dear Dr. Siame, 

I'm pleased to inform you that your manuscript has been deemed suitable for publication in PLOS ONE. Congratulations! Your manuscript is now being handed over to our production team.

Kind regards, 

on behalf of

Dr. Rayan Jafnan Alharbi 

Academic Editor

PLOS ONE